# Biopsy Confirmed Glioma Recurrence Predicted by Multi-Modal Neuroimaging Metrics

**DOI:** 10.3390/jcm8091287

**Published:** 2019-08-23

**Authors:** Jamie D. Costabile, John A. Thompson, Elsa Alaswad, D. Ryan Ormond

**Affiliations:** 1Department of Neurosurgery, University of Colorado School of Medicine, Aurora, CO 80045, USA; 2Department of Neurology, University of Colorado School of Medicine, Aurora, CO 80045, USA

**Keywords:** glioma, diffusion tensor imaging, generalized q-ball imaging, treatment-related effects, multiple resections

## Abstract

Histopathological verification is currently required to differentiate tumor recurrence from treatment effects related to adjuvant therapy in patients with glioma. To bypass the complications associated with collecting neural tissue samples, non-invasive classification methods are needed to alleviate the burden on patients while providing vital information to clinicians. However, uncertainty remains as to which tissue features on magnetic resonance imaging (MRI) are useful. The primary objective of this study was to quantitatively assess the reliability of combining MRI and diffusion tensor imaging metrics to discriminate between tumor recurrence and treatment effects in histopathologically identified biopsy samples. Additionally, this study investigates the noise adjuvant radiation therapy introduces when discriminating between tissue types. In a sample of 41 biopsy specimens, from a total of 10 patients, we derived region-of-interest samples from MRI data in the ipsilateral hemisphere that encompassed biopsies obtained during resective surgery. This study compares normalized intensity values across histopathology classifications and contralesional volumes reflected across the midline. Radiation makes noninvasive differentiation of abnormal-nontumor tissue to tumor recurrence much more difficult. This is because radiation exhibits opposing behavior on key MRI modalities: specifically, on post-contrast T1, FLAIR, and GFA. While radiation makes noninvasive differentiation of tumor recurrence more difficult, using a novel analysis of combined MRI metrics combined with clinical annotation and histopathological correlation, we observed that it is possible to successfully differentiate tumor tissue from other tissue types. Additional work will be required to expand upon these findings.

## 1. Introduction

An important challenge facing the neuro-oncological treatment of gliomas is discriminating between tumor recurrence and treatment-related effects using non-invasive diagnostic imaging [1]. Not only do tissue types appear similar on standard magnetic resonance imaging (MRI), but new lesions are often a composite of tumor cells, gliosis, necrosis, inflammatory cells, and neovascularity, which confounds characterization [2]. Moreover, targeted therapies like bevacizumab complicate follow-up imaging even further by modifying vascular endothelial growth factor (VEGF), often causing a “pseudoresponse” with vascular changes resulting in a subsequent decrease in contrast enhancement [3]. Similarly, changes related to radiation or immunotherapy can mimic tumor progression, including changes in T1-weighted (T1w) contrast enhancement and T2-weighted (T2w) hyperintensity, once again complicating imaging-based tissue discrimination [4]. Etiological characterization of lesions observed on longitudinal follow-up scans factors into the clinical decision-making in the course of treatment and prognostic decisions.

While histopathology remains the gold standard for tissue type identification, it is not without its problems, such as the need for additional surgery, sampling bias, and risks of neurological complication [2,5]. Thus, a non-invasive method capable of distinguishing recurrence from treatment effects must be established in order to reduce the dependency on biopsy and improve the efficacy of patient follow-up with noninvasive imaging. Advanced MRI methods such as magnetic resonance (MR) spectroscopy, MR perfusion, positron emission tomography (PET), single photon emission CT (SPECT), diffusion weighted imaging (DWI), and diffusion tensor imaging (DTI) have been used to explore the feasibility of differentiating tumor recurrence and treatment effects with varying success [6,7,8,9,10,11]. PET-based methods, which measure glucose metabolism, demonstrate some ability in distinguishing glioma recurrence from radiation-induced necrosis. For example, increased fludeoxyglucose (FDG) tracer activity, corresponding to enhanced uptake on post-contrast T1 imaging, is consistent with tumor recurrence, while decreased FDG tracer activity is less specific, typically denoting vasogenic edema, stemming from recurrence and treatment effects [12,13]. Amino acid transport PET-based imaging, especially the use of tyrosine or tryptophan-based tracers, has also been studied to improve the ability to distinguish tumor recurrence from treatment-related changes. O-(2-[18F]fluoroethyl)-L-tyrosine (FET) has been studied since the 1990s and is believed to be more specific for tumor recurrence given the enhanced uptake of glucose in all brain (FDG) versus less amino acids uptake [14]. This should make FET PET more specific than FDG, and there have been a number of cases showing increased uptake of FET in tumors, and it may also be useful at assessing pseudoprogression from true recurrence in glioma [15,16,17,18,19,20]. However, several other tissue types can also have increased uptake, including brain abscesses, demyelinating processes, epilepsy, and in tissue adjacent to cerebral ischemia or hematomas, making some interpretation of results challenging [14,15,17]. MR perfusion techniques, like dynamic contrast-enhanced (DCE) MRI and dynamic susceptibility contrast (DSC) MRI, yield estimates of relative cerebral blood volume (rCBV) and vascular permeability (k_trans_), reflecting underlying microvasculature and angiogenesis [21,22,23]. Studies have indicated MR perfusion’s utility in differentiating tumor progression from treatment effects and pseudoprogression [24,25,26]. However, these techniques are hindered by mixed results [27], model complexity [28], and sensitivity to thresholds [29]. MR spectroscopy, estimating biomarkers like lactate and choline to creatinine ratios, has demonstrated higher diagnostic accuracy than conventional MRI in detecting tumor progression as well, reaching a sensitivity and specificity as high as 91% and 95%, respectively [30]. The diffusion metrics fractional anisotropy (FA) and mean diffusivity (MD) have been useful in differentiating between tissues types as well [31,32,33]. Recent research on glioblastoma demonstrated that MD can help differentiate between tumor recurrence and radiation-induced necrosis, as it is known that more free water lies within necrotic tissue than enhancing solid tumor [34]. Also, Apparent Diffusion Coefficient (ADC) ratios and mean ADC of tumor recurrence are significantly lower than those of radionecrosis, since higher cellularity (tumor recurrence) contributes to more restricted diffusion [35]. Verma et al. (2013) suggests the combination of low ADC values and high FA values help define the presence of tumor recurrence [2].

High grade gliomas, the most prevalent intracranial neoplasm, are highly heterogenous in the lesion area, have an invasive nature, and often require additional multimodality treatment later in the course of the disease. For these reasons, noninvasive diagnosis, monitoring, and prognosis strategies, such as MRI, must be sought and refined. With the goal to improve the noninvasive diagnostic utility of advanced MRI for gliomas, we studied a group of patients who had imaging localized histopathology. Through the combination of both conventional and advanced MRI modalities, we demonstrate improved efficacy in diagnosing recurrent tumor versus imaging effects related to treatment. These results demonstrate the potential for refining multi-modal MRI assessment of glioma tissue classification, thereby facilitating the clinical decision-making process.

## 2. Experimental Section

### 2.1. Patient Information

All procedures and protocols for this study were reviewed and approved by the Colorado Multi-Institutional Review Board (COMIRB 17-1136). Subjects included in this study were patients undergoing repeat resective surgery after radiologically defined tumor progression between August and November 2018 at the University of Colorado Hospital. The patient set consisted of 10 subjects who received prior resection(s) for recurrent glioma with detailed histopathology recorded for 2 or more biopsies (41 biopsies collected in total). Data were collected retrospectively from patient chart review. Two patients received two prior resections; all others received one prior resection. The patient set is divided into two groups: those that underwent radiation therapy prior to repeat resection (RT, *n* = 7) and those that did not (No RT, *n* = 3). For each patient, biopsy samples were collected during surgery from the radiologically-defined tumor region and examined by an expert neuropathologist (B.K.D.). The neuropathologist classified each sample and an expert neurosurgeon (D.R.O.) designated each classification as primarily consisting of abnormal, nontumor tissue (Abnormal), or tumor tissue (Tumor). Patient information is summarized in Table 1.

### 2.2. Imaging Sequence Parameters

All images were obtained using a 3.0-T whole-body MR imager (Signa HDxt; GE Medical Systems, Milwaukee, Wisconsin, USA) between 0–24 days prior to repeat surgical intervention. Acquisition times were 2.5, 5.4, 4.6, 7.8, and 9.0 minutes for non-enhanced T1-weighted (T1w), gadolinium-enhanced T1-weighted (T1ce), T2-weighted (T2w), T2-FLAIR (FLAIR), and diffusion-weighted (DW) images, respectively. For T1w, TE = 2.3 ms, TR = 5.5 ms, and flip angle = 8°. Data were recorded as a 256 × 256 matrix with 1 mm × 1 mm pixel spacing, a slice thickness of 1.2 mm, and zero slice gap. For T1ce, TE = 2.5 ms, TR = 6.8 ms, and flip angle = 8°. Data were recorded as a 512 × 512 matrix with 0.5 mm × 0.5 mm pixel spacing, a slice thickness of 1.2 mm, and zero slice gap. For T2w, TE = 6333 ms, TR = 80 ms, and flip angle = 142°. Data were recorded as a 512 × 512 matrix with 0.5 mm × 0.5 mm pixel spacing, a slice thickness of 2 mm, and zero slice gap. For FLAIR, TE = 6000 ms, TR = 128 ms, and flip angle = 90°. Data were recorded as a 512 × 512 matrix with 0.5 mm × 0.5 mm pixel spacing, a slice thickness of 1.2 mm, and zero slice gap. For DW images, TE = 85 ms, TR = 16,000 ms, and flip angle = 90°. The diffusion gradient was encoded in 32 directions at b = 1000 s/mm^2^ and an additional measurement without the diffusion gradient (b = 0 s/mm^2^). DW data were recorded as a 128 × 128 matrix with 0.9375 mm × 0.9375 mm pixel spacing. A total of 50 sections were obtained with a slice thickness of 2.6 mm and zero slice gap.

### 2.3. Image Processing

Images were processed using a combination of open-source software packages: MRtrix [36], FSL [37], and greedy [38]. Standard MR images (T1w, T1ce, T2w, and FLAIR) were non-linearly registered to the MNI152 (Montreal Neurological Institute, MNI) atlas [39] space using the deformable registration package greedy. Automated tissue-type segmentation was performed on T1w image sets using FSL-FAST (FMRIB’s Automated Segmentation Tool) [40]. DT images were preprocessed to remove noise and corrected for distortion and field-bias using MRtrix’s dwidenoise [41], dwipreproc [42], and dwibiascorrect [40,43] scripts. After preprocessing, DT images were linearly registered into T1w-space using FSL-FLIRT (FMRIB’s Linear Image Registration Tool) [44] and then transformed into MNI-space by applying the affine matrix generated to register the T1w image. Lastly, all image sets were downsampled by a factor of 0.45 with cubic interpolation using MRtrix to avoid oversampling (voxel size: 1.75 mm^3^).

### 2.4. Image Normalization

MR image intensities are acquired in arbitrary units, introducing noise when comparing scans taken at different times. To compensate for artifacts between scans, each MR and DW sequence were normalized across the patient set. Standard MR sequences were normalized using the RAVEL method [45] implemented with the intensity-normalization library [46]. The DW sequence was normalized using MRtrix’s dwiintensitynorm.

### 2.5. Diffusion Feature Space

All diffusion features were calculated using DSI Studio (http://dsi-studio.labsolver.org) on processed and normalized diffusion-weighted images. The diffusion information was reconstructed in two fashions using diffusion tensor [47] and generalized q-space imaging [48]. Diffusion tensor imaging (DTI) determines three primary diffusion directions (and magnitudes) using a tensor, from which the standard diffusion metrics fractional anisotropy (FA) and mean diffusivity (MD) were determined. Generalized q-ball imaging (GQI) is a model-free method that calculates the orientation distribution of the density of diffusing water. Using GQI, the non-standard diffusion metrics quantitative anisotropy (QA) and generalized fractional anisotropy (GFA) were determined. A diffusion sampling length ratio of 1.25 was used. The b-table was checked by an automatic quality control routine to ensure its accuracy [49]. Diffusion feature (FA, MD, QA, and GFA) maps were extracted for each subject from normalized diffusion images.

### 2.6. Regions of Interest (ROI)

During resective surgery, the locations of biopsies on the patient’s MRI were identified using a Medtronic StealthStation S8 Surgical Navigation system (Medtronic, Minneapolis, MN, USA) and application software (Version 1.1.0-39). The biopsy locations were recorded via screenshots. With this information, voxel locations were manually identified on our analytical setup and transformed into MNI-space by applying the patient’s transformation affine. A one-half cubic centimeter sphere was used as a facsimile for the biopsy in MR image space.

### 2.7. Data Analysis

All data analysis was performed using the programming language Python with NiBabel, Numpy, Pandas, Seaborn, Scipy, and Statsmodels modules.

## 3. Results

### 3.1. Image Analysis of Biopsy Classifications

The image data analyzed in this study is summarized in Figure 1. Eight MR/DW image features (T1w, T1ce, T2w, FLAIR FA, MD, QA, and GFA) were collected from each patient prior to re-resection. Each image feature was normalized across patients to account for fluctuations in signal acquisition due to environmental and equipment variations (Figure 1A). The image intensities were extracted from ROIs representing the locations of surgical biopsies along with their contralaterally Normal analogs (Figure 1B). Example photomicrographs of the Abnormal and Tumor biopsy classifications from one patient are displayed in Figure 1C.

To explore the effect of radiation therapy on biopsy classification, mean signal intensities were calculated for each ROI and separated based on treatment group (Figure 2). For No RT patients (Figure 2A), differences were detected between Abnormal and Tumor in the T1ce and T2w signals (Tukey’s post-hoc test, Family Wise Error Rate (FWER) = 0.05) and between Tumor and Normal in the T1w, T1ce, T2w, FLAIR, FA, and MD signals (Tukey’s post-hoc test, FWER = 0.05). No differences were detected between Abnormal and Normal. For RT patients (Figure 2B), fewer image features were deemed statistically different. No differences were detected between Abnormal and Tumor (Tukey’s post-hoc test, FWER = 0.05), one difference was detected between Tumor and Normal in the T1ce signal (Tukey’s post-hoc test, FWER = 0.05), and two differences were detected between Abnormal and Normal in FLAIR and MD signals. The only difference consistent among treatment groups was between Tumor and Normal for the T1ce image modality; though, the feature demonstrated a reversed behavior between the two groups. Mean Normal signal intensities were equal between groups in all MRI modalities excluding MD (Appendix A).

### 3.2. Logistic Regression Modeling

Given the overall lack of consensus for features that consistently discriminated between treatment groups, we evaluated the ROI image intensities on the voxel-level (Figure 3A,B) to the presence of Tumor (Figure 3C,D) using logistic regression. The regression coefficients provide an estimate of the explained variance each image modality has on the likelihood of the presence of Tumor. Models incorporating all eight image features were created for each treatment and the resulting regression coefficients were calculated (Figure 3C). The significant features consistent in both models were T1ce, FLAIR, QA, and GFA (Student’s t test, corrected for multiple comparisons using False Discovery Rate (FDR), *p* < 0.05). However, T1ce, FLAIR, and GFA express inverted information between the models: T1ce shows that for the RT group, higher intensities indicated the presence of Tumor tissue, whereas for the No RT group, higher intensities indicated the presence of Abnormal tissue. The converse is true for FLAIR and GFA: for the RT group, higher intensities indicated the presence of Abnormal tissue, and for the No RT group, higher intensities indicated the presence of Tumor tissue. Therefore, the same approach for differentiating Abnormal and Tumor tissue for patients in the No RT group is not wholly applicable to patients in the RT group (only for QA). Figure 3D illustrates how the No RT and RT models—built using the T1ce, FLAIR, QA, and GFA features—perform similarly (area under curve (AUC) = 0.84 and AUC = 0.75, respectively) when accounting for treatment. However, the aggregate model (“All patients”, Figure 3D) performed the worst (AUC = 0.60)—showing that the conflicting information (demonstrated in Figure 2 and Figure 3C) degraded the model’s ability to differentiate Abnormal and Tumor tissue using multi-modal MRI.

## 4. Discussion

In order to more specifically evaluate imaging changes consistent with treatment-related effects versus tumor recurrence, we began collecting voxel-based MRI information coupled with location specific blinded histopathological review using a within subject experimental design (i.e., contralesional matched normal voxel as a normal brain control). The goal of this project was to ultimately identify hurdles in predictive modeling regarding imaging diagnoses when longitudinally following patients with glioma after treatment to better assess true recurrence when MR changes occur, incorporating the use of DTI into standard algorithms. Frequently, changes occur on MRI after treatment, which can be difficult to interpret. Treatments such as immunotherapy (still experimental), radiation, or cytotoxic therapy often induce changes in T2w hyperintensity and T1w contrast enhancement that can occur even several years after treatment has ended [4,50,51,52]. Additionally, targeted therapies, such as bevacizumab, can decrease contrast enhancement and hyperintensity, sometimes masking progression [3]. These challenges in imaging interpretation have been well known for many years. Defining progression in glioma has always been difficult and somewhat controversial.

First described by Macdonald et al., in 1990, the Macdonald criteria were imaging-based criteria to determine glioma progression based on contrast enhancement in two dimensions on CT scans in patients undergoing treatment [53]. This was later adapted to MRI and included four response categories: complete response, partial response, stable disease, or progressive disease. Macdonald criteria is limited by irregularly shaped tumors or nonspecific contrast enhancement from pharmacological treatments, radiation, inflammation, necrosis, pseudoprogression, etc. [54,55,56]. It also does not account for noncontrast enhancing disease, which is especially important in the evaluation of diffuse low-grade glioma. In 2010, the RANO Criteria consortium published, and later modified, guidelines for the evaluation of treatment response in gliomas and incorporated nonspecific contrast enhancement, multifocal tumors, pseudo-response after treatment, and nonenhancing fluid-attenuated inversion-recovery (FLAIR) hyperintense region in determining treatment response [57,58]. More recent measures of clinical progression have been developed to also help in distinguishing between true progression and pseudoprogression [59,60]. While these measures are important in assessing the global status of the patient and are quite sensitive and specific for global tumor recurrence, they do not answer the challenge of voxel-by-voxel analysis of imaging features specific for tumor recurrence. This study helps to further efforts of predictive, noninvasive modeling by investigating chemoradiation therapy influence on imaging in the process of determining tumor recurrence. These models can also be used to potentially better predict presence of residual disease following surgery, sites of future disease progression, and progression free survival.

This study investigated the effects of surgery alone or surgery plus radiation on voxel-specific pathology. Overall, radiation makes noninvasive differentiation of abnormal-nontumor tissue to tumor recurrence much more difficult. This is because radiation exhibits opposing behavior on key MRI modalities: specifically, on post-contrast T1, FLAIR, and GFA (a GQI feature related to FA). A number of treatment modalities clearly distinguish tumor from abnormal-nontumor postoperatively, however many of these features lose their distinguishing characteristics after radiation (see Figure 2). Specifically, features significant in both models (T1ce, FLAIR, and GFA) demonstrate contrasting information dependent on the postsurgical treatment strategy. T1ce shows that for the RT group, higher intensities indicate the presence of tumor where for the No RT group, higher intensities indicate the presence of abnormal tissue not containing tumor. The converse is true for FLAIR and GFA: for the RT group, higher intensities indicate the presence of abnormal, nontumor tissue, while lower intensities indicate tumor tissue (see Figure 3C). This implies that in order to differentiate abnormal-nontumor tissue from tumor tissue, understanding previous treatment modalities is imperative. The same approach for discriminating one for the other will not work depending on prior treatment.

Violin plots of standard MRI features (Figure 3A,B) help to understand these shifts in a more granular way. Shifts in the histograms happen all along normalized intensity values with nearly all features tested. This is predictable and influenced by treatment strategy, although histograms appear more similar after radiation, demonstrating the difficulty of distinguishing recurrence from post-treatment effects after radiation using standard features of MRI. Standard measurements also differed significantly from normal with or without radiation (Figure 2). However, distinguishing between tumor and abnormal-nontumor was difficult. FA and MD, specifically, provided no information to distinguish tumor from abnormal-nontumor tissue after radiation, although QA and GFA did. Instead, logistic regression helped to illustrate which features contributed most to differentiating the biopsy labels of tumor versus abnormal-nontumor. Hence, the opposing but important findings described previously of T1ce, FLAIR, and GFA and their predictive value in our model. Whereas, QA (quantifies the spin orientation population in a specific direction) remained consistent across treatment models. Ultimately, both models separating images by prior treatment modality (both groups had prior surgery, some with or without chemoradiation prior to re-resection) performed well, while the aggregate model “all patients” performed poorly. This shows that the conflicting information demonstrated in Figure 3C degrades the model’s ability to differentiate abnormal-nontumor from tumor tissue on MRI unless separated by treatment modality.

Overall, including non-standard DTI metrics is a useful addition towards differentiation between tumor recurrence and abnormal-nontumor MRI changes, although more is needed in the effort to improve accurate noninvasive prediction of recurrence. This study demonstrates the continued importance of matching imaging data to pathology and clinical annotation to avoid misinterpreting findings on MRI. Ultimately, combining complex datasets including pathology, genomics, epigenetics, imaging, and clinical information will all be important in improving noninvasive assessment of glioma. Future studies including more patients and more precise imaging/pathology correlation will help improve our predictive modeling to the betterment of the care of glioma patients.

## 5. Conclusions

Radiation makes the noninvasive differentiation of abnormal-nontumor tissue vs tumor recurrence much more difficult. This is because radiation exhibits opposing behavior on key MRI modalities: specifically, on post-contrast T1, FLAIR, and GFA. Ultimately, combining multiple MRI metrics with clinical annotation allows the more successful differentiation of tumor recurrence from other post-treatment effects on MRI. 

## Figures and Tables

**Figure 1 jcm-08-01287-f001:**
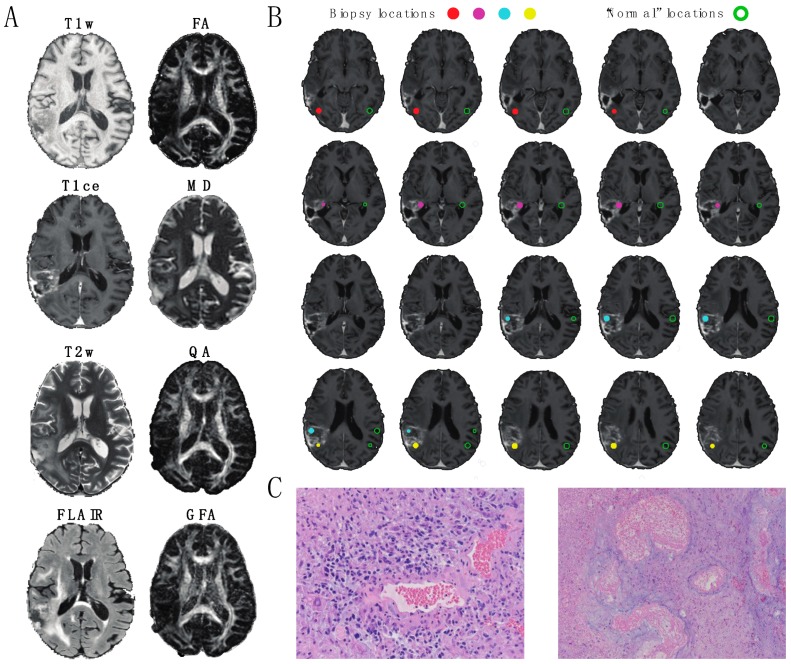
A 59-year old male patient with glioblastoma multiforme. (**A**) Axial slices of the image modalities explored in this study, comprised of four standard MRI metrics (T1w, T1ce, T2w, FLAIR = fluid-attenuated inversion recovery) and four diffusion MRI metrics (fractional anisotropy (FA) and mean diffusivity (MD) quantitative anisotropy (QA) and generalized fractional anisotropy (GFA)). (**B**) Depiction of biopsies from the patient shown in (A). Filled circles indicate the locations of 0.5 mm^3^ Regions of Interest (ROIs) representing tissue extractions. Open circles indicate the locations of anatomically similar locations of 0.5 mm^3^ ROIs in the normal appearing (“healthy”) contralateral hemisphere. For this patient, one biopsy (red) consisted primarily of abnormal tissue and three biopsies (magenta, cyan, and yellow) consisted primarily of tumor tissue. (**C**) Example slides of histopathology used in classification. (Left image) Tumor: Infiltrating high-grade glioma is seen with cytologically pleomorphic nuclei with large areas of necrosis and thick hyalinized blood vessels (20× magnification). (Right image) Abnormal: cortical white matter with extensive gliosis and neuropil vacuolization. Regional necrosis with thick hyalinized blood vessels consistent with radiation necrosis is present (10× magnification).

**Figure 2 jcm-08-01287-f002:**
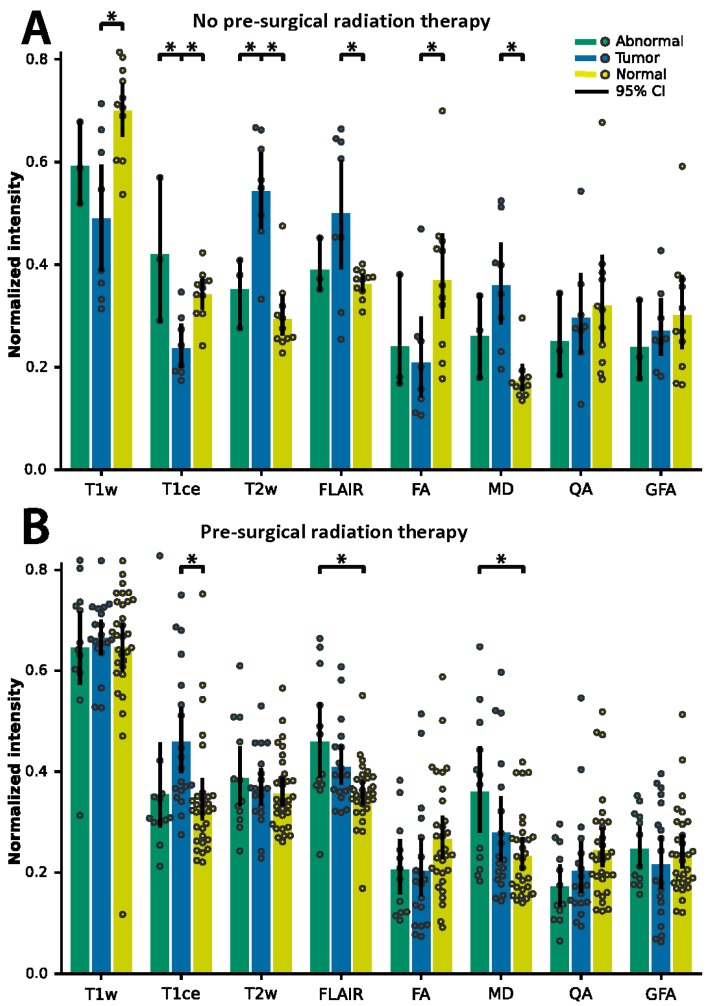
Average ROI normalized image intensity for biopsies classified as Abnormal (green) and Tumor (blue). Contralaterally mirrored ROI locations classified as Normal (yellow). Data separated depending on chemoradiation therapy strategy prior to re-resection: (**A**) patients with adjuvant radiation therapy and (**B**) patients without adjuvant radiation therapy. Error bars show 95% confidence intervals. Asterisks indicate significance determined using Tukey’s post-hoc test, *p* < 0.05.

**Figure 3 jcm-08-01287-f003:**
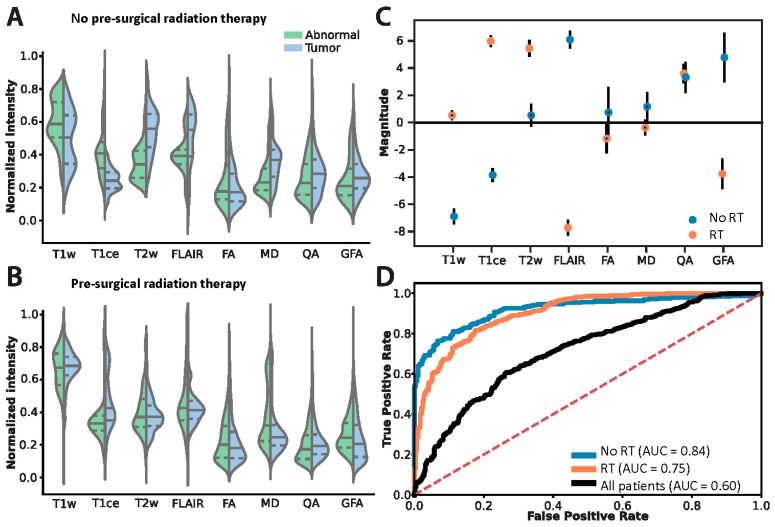
Differentiating the histopathology classifications Abnormal and Tumor on the voxel-level accounting for prior chemoradiation treatment regime. Voxel intensity histograms from the (**A**) No RT and (**B**) RT groups. Solid lines indicate median, dashed lines indicate the lower and upper interquartile interval. (**C**) Logistic regression coefficients: filled circles indicate significant features in the model, open circles indicate non-significant features. Error bars show standard deviation. (**D**) Logistic regression model performance using only the features deemed significant in (**C**). ROC denotes “receiver operator characteristics”, AUC denotes “area under curve”, and the “All patients” model (built only using features significant in both models) is an aggregate of the treatment groups.

**Table 1 jcm-08-01287-t001:** Clinical data of the patient set.

Age	Sex	Location & Pathology	IDH/MGMT/EGFR Status	Time between Imaging and Surgery (Days)	Months Since Prior Resection	RT Prior to Latest Resection	CT prior to Latest Resection	No. of Abnormal Biopsies	No. of Tumor Biopsies
59	M	Right occipital, glioblastoma multiforme	WT/−/lo	2	4.0	Yes	Yes	1	3
34	F	Left frontal, diffuse astrocytoma	MT/NA/NA	2	14.8	No	No	0	4
32	M	Left frontal, anaplastic oligodendroglioma	MT/NA/NA	0	70.4	Yes	Yes	0	4
62	M	Right temporal, glioblastoma multiforme	WT/+/moderate	4	49.2	Yes	Yes	4	1
36	F	Right frontal, glioblastoma multiforme	MT/−/No/BRAF V600E mut	24	27.7	Yes	Yes	6	0
32	M	Right frontal, glioblastoma multiforme	MT/NA/neg	7	62.6	Yes	Yes	0	4
32	F	Right frontal, oligodendroglioma	MT/NA/NA	7	21.5	No	Yes	1	4
58	M	Right tempoparietal, glioblastoma multiforme	WT/+/hi	2	2.8	Yes	Yes	1	2
31	M	Right frontal, diffuse astrocytoma	MT/−/lo	0	51.6	No	No	2	0
42	M	Right frontal, glioblastoma multiforme	WT/NA/lo	10	26.0	Yes	Yes	0	4

Abbreviations: M = male, F = female, MT = mutant, WT = wild type, NA = not available, lo = low expression, hi = high expression, + = methylated, − = unmethylated, IDH = isocitrate dehydrogenase, MGMT = O-6-methylguanine-DNA-methyltransferase, EGFR = epidermal growth factor receptor, BRAF = v-Raf murine sarcoma viral oncogene homolog B, RT = radiation therapy, CT = chemotherapy.

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
