# Peer review of "Biopsy Confirmed Glioma Recurrence Predicted by Multi-Modal Neuroimaging Metrics"

_jcm, 2019, doi:10.3390/jcm8091287_

Round 1

Reviewer 1 Report

The study explores an important and challenging question in management of glioma patients, namely differentiation of tumor tissue from treatment related changes. The paper is well written and analyses are clearly presented.

Please find my suggestions below:

Please specify molecular biomarkers (if available), such as MGMT, IDH status, as well as chemotherapy and radiotherapy previous treatments.

Time window from MRI to biopsy can affect intraoperative targeting. Please specify.

Could authors specify in more detail how the imaging index based on regression coefficients was established?

Do authors have information about reliability of commonly used imaging techniques namely T1w contrast enhanced and T2 to predict presence/absence of tumor tissue during biopsies as these methods remain the most commonly used for image guidance during tumor resection surgery.

Author Response

Please specify molecular biomarkers (if available), such as MGMT, IDH status, as well as chemotherapy and radiotherapy previous treatments.

We have added MGMT, IDH or other molecular biomarker status, as appropriate, and when available, to Table 1.

Time window from MRI to biopsy can affect intraoperative targeting. Please specify.

Imaging was performed close to the time of surgery where specimen was obtained. Days prior to surgery for imaging is now included in Table 1.

Could authors specify in more detail how the imaging index based on regression coefficients was established?

We have added additional details to the model description to clarify how the imaging indices were incorporated into the regression coefficients. See lines 214-216.

Do authors have information about reliability of commonly used imaging techniques namely T1w contrast enhanced and T2 to predict presence/absence of tumor tissue during biopsies as these methods remain the most commonly used for image guidance during tumor resection surgery.

The reviewer is correct to point out that most assessments rely on these image modalities for convenience. However, the goal of this manuscript was to objectively and quantitatively assess whether these and other image modalities could differentiate between recurrence and abnormal tissue. It has been well-documented that the utility and reliability of T1w and T2-contrast enhanced imaging alone to assess evaluate gliomas is insufficient.

Reviewer 2 Report

The authors of, "Biopsy confirmed glioma recurrence predicted by multi-modal neuroimaging metrics" have completed a small study comparing patient tumor tissues and clinical imaging techniques to assess the potential for the imaging to predict tissue histology. A non-invasive imaging technique to distinguish between tumor recurrence and treatment effects would invaluable to advise physicians how to treat their patients. 

Despite the small sample size, this study would be valuable to other glioma researchers and physicians. 

The only minor concern is that in the introduction and discussion, the authors have failed to include any information about amino acid PET imaging. Radiotracers based on tyrosine or tryptophan have been demonstrated in the literature to be able to predict tumor recurrence vs. radiation necrosis and are under development for clinical use. Inclusion of amino acid PET tracers would provide a more complete survey of neuroimaging. 

Author Response

Despite the small sample size, this study would be valuable to other glioma researchers and physicians.

We thank the reviewer for their comments on the manuscript quality.

The only minor concern is that in the introduction and discussion, the authors have failed to include any information about amino acid PET imaging. Radiotracers based on tyrosine or tryptophan have been demonstrated in the literature to be able to predict tumor recurrence vs. radiation necrosis and are under development for clinical use. Inclusion of amino acid PET tracers would provide a more complete survey of neuroimaging.

Amino acid-based PET imaging is under development for clinical use to help predict tumor recurrence vs radiation necrosis. A brief discussion of this development is now included in the introduction. See Lines 59-66.